# Associations of Multimarkers of Metabolic Malnutrition and Inflammation with All-Cause Mortality by Multimorbidity Status

**DOI:** 10.3390/nu17111747

**Published:** 2025-05-22

**Authors:** Setor K. Kunutsor, Reyhaneh Rikhtehgaran, Margery A. Connelly, Irina Shalaurova, Stephan J. L. Bakker, Robin P. F. Dullaart

**Affiliations:** 1Section of Cardiology, Department of Internal Medicine, Rady Faculty of Health Sciences, University of Manitoba, Winnipeg, MB R2H 2A6, Canada; reyhaneh.rikhtehgaran@umanitoba.ca; 2Labcorp, Morrisville, NC 27560, USA; connem5@labcorp.com (M.A.C.); shalaui@labcorp.com (I.S.); 3Department of Internal Medicine, Division of Nephrology, University Medical Center Groningen, University of Groningen, 9700 RB Groningen, The Netherlands; s.j.l.bakker@umcg.nl; 4Department of Internal Medicine, Division of Endocrinology, University Medical Center Groningen, University of Groningen, 9700 RB Groningen, The Netherlands; dull.fam@12move.nl

**Keywords:** inflammation vulnerability index, multimorbidity, mortality, cohort study

## Abstract

**Background/Objectives**: The metabolic vulnerability index (MVX)—a composite biomarker reflecting metabolic malnutrition and inflammation—is associated with increased mortality risk, but its association across different levels of chronic disease burden has not been explored. We aimed to examine the associations of MVX and its subcomponents (Inflammation Vulnerability Index, IVX and Metabolic Malnutrition Index, MMX) with all-cause mortality according to multimorbidity status. **Methods**: In the PREVEND study, which included 6054 participants (mean age 54 years; 49.5% male), MVX was calculated using six plasma biomarkers measured simultaneously via nuclear magnetic resonance spectroscopy. Hazard ratios (HRs) with 95% confidence intervals (CIs) were estimated. **Results**: During a median follow-up of 14.0 years, 911 deaths were recorded. In analyses adjusted for several established risk factors, the HRs (95% CIs) of mortality per 1 standard deviation increment in MVX were 1.32 (1.13–1.54; *p* < 0.001), 1.23 (1.08–1.40; *p* = 0.002), and 1.29 (1.16–1.43; *p* < 0.001) for individuals with no disease, one disease, and multimorbidity, respectively. The corresponding HRs (95% CIs) were 1.22 (1.05–1.42; *p* = 0.010), 1.17 (1.03–1.34; *p* = 0.015), and 1.25 (1.13–1.38; *p* < 0.001) for IVX and 1.29 (1.11–1.48; *p* = 0.001), 1.16 (1.02–1.31; *p* = 0.032), and 1.14 (1.03–1.25; *p* = 0.004) for MMX. The ratio of HRs showed no statistical evidence that sex modified the associations of MVX, IVX, and MMX with mortality in each multimorbidity category. However, the associations appeared stronger in males with chronic disease and in females without chronic conditions, suggesting possible sex-related trends. **Conclusions:** MVX, IVX, and MMX are independent risk indicators of all-cause mortality regardless of the burden of chronic disease, with MVX showing the strongest associations across different multimorbidity statuses. MMX should be interpreted as a proxy for metabolic malnutrition rather than a direct nutritional assessment tool.

## 1. Introduction

Multimorbidity, defined as the coexistence of two or more chronic conditions in an individual [1], is a growing public health challenge that is associated with increased healthcare utilization, reduced quality of life, and an increased risk of mortality [2,3,4]. As the number of chronic diseases increases, the physiological burden intensifies, exacerbating metabolic and inflammatory dysfunctions that contribute to adverse health outcomes. Individuals with multimorbidity are particularly vulnerable, with heightened mortality risk as the burden of chronic disease grows [5,6]. Despite the well-established link between multimorbidity and mortality, traditional risk markers have limited utility in capturing the complex interplay between metabolic and inflammatory dysregulations that underlie these associations.

Recent research has introduced the Metabolic Vulnerability Index (MVX), a composite score derived from six plasma biomarkers, as a strong predictor of mortality [7]. MVX integrates markers of systemic inflammation and metabolic malnutrition, which are central to conditions such as cachexia, sarcopenia, and frailty—syndromes frequently seen in individuals with chronic diseases and associated with poor survival outcomes [7,8]. MVX consists of two key subcomponents: the Inflammation Vulnerability Index (IVX), which captures systemic inflammation through markers such as GlycA and small high-density lipoprotein particles (S-HDL-P), and the Metabolic Malnutrition Index (MMX), which reflects protein-energy wasting using metabolites such as valine, leucine, isoleucine, and citrate [7]. These multimarkers can offer a more comprehensive understanding of mortality risk, particularly when stratified by the burden of chronic disease.

Previous research has demonstrated the utility of MVX in populations at high risk of cardiovascular disease (CVD), by showing that it is a strong predictor of mortality that is independent of conventional cardiometabolic risk factors [7,8]. However, its associations with mortality risk across different levels of chronic disease burden remain underexplored. Given that metabolic malnutrition and inflammation may play universal roles in shaping long-term survival [9,10], further investigation is warranted to determine how these indices perform in diverse populations. This study seeks to expand on prior research by investigating the associations of MVX, IVX, and MMX with all-cause mortality based on chronic disease burden, specifically among individuals with no disease, one disease, or multimorbidity. The Prevention of Renal and Vascular End-stage Disease (PREVEND) study, a well-characterized population-based observational cohort, provides an ideal setting for this investigation. This study aimed to examine the associations of MVX and its subcomponents (IVX and MMX) with all-cause mortality according to multimorbidity status (presence of no disease, one chronic disease, or multimorbidity). Understanding these relationships could enhance risk stratification and inform targeted interventions aimed at reducing the risk of mortality in populations with varying levels of chronic disease burden.

## 2. Materials and Methods

### 2.1. Study Design and Population

The study followed the STROBE (STrengthening the Reporting of Observational Studies in Epidemiology) guidelines for observational research (Appendix A) [11]. The data were obtained from the PREVEND study, a well-established Dutch cohort designed to examine the prognostic value of urinary albumin excretion in relation to renal and cardiovascular health outcomes. Detailed descriptions of the cohort’s design, participant selection, and data collection procedures are available in earlier publications [12]. Ethical approval for the PREVEND study was granted by the Medical Ethics Committee of the University Medical Center Groningen (approval code MEC 96/01/022, approval date 25 March 1996), and the study was conducted in accordance with the Declaration of Helsinki. Written informed consent was obtained from all participants prior to their enrollment. The cohort was drawn from the general adult population of Groningen, the Netherlands, with recruitment strategies designed to ensure demographic representativeness. In the initial screening phase (1997–1998), 8592 individuals completed baseline assessments. This was followed by a second screening phase (2001–2003), which serves as the baseline for the present analysis and included 6894 individuals aged 32 to 80 years. For the current study, we included 6054 participants who had complete data available on the metabolic vulnerability markers (MVX, IVX, and MMX), multimorbidity classification, relevant covariates, and mortality outcomes, as detailed in Appendix A.

### 2.2. Assessment of Exposures, Additional Risk Markers and Multimorbidity Status

Data for this study were collected during two visits at an outpatient clinic, where demographics, lifestyle behaviors, medical history, physical characteristics, and medication use were recorded. Blood pressure was measured multiple times at each visit, with the final value calculated as the average of the last two readings. Type 2 diabetes (T2D) was identified based on fasting plasma glucose (FPG) ≥ 7.0 mmol/L (126 mg/dL), random plasma glucose ≥ 11.1 mmol/L (200 mg/dL), self-reported physician diagnosis, or the use of glucose-lowering medication recorded in a centralized pharmacy database. Ref. [13] Body mass index (BMI) was derived as weight (kg) divided by height squared (m^2^), and alcohol intake was assessed via self-reported questionnaires. Venous blood samples were drawn after an overnight fast and analyzed for biochemical markers. EDTA-anticoagulated plasma samples were stored at <−70 °C before being shipped to Labcorp (Morrisville, NC, USA), where metabolic markers were quantified using nuclear magnetic resonance (NMR) spectroscopy on a Vantera^®^ Clinical Analyzer (Morrisville, NC, USA). The algorithm that reports MVX, IVX, and MMX scores was developed from cohorts at high cardiovascular risk and the equations have been previously reported [7,14]. The IVX, a key component of MVX, was calculated based on S-HDL-P and GlycA levels, with GlycA reflecting the quantity of *N*-acetyl methyl group protons on acute phase glycoproteins that circulate at fairly high concentrations. The MMX was determined using citrate and the branched-chain amino acids (valine, leucine, and isoleucine). The MVX score was computed by integrating IVX and MMX scores, with all indices scaled from 1 to 100, where higher values indicate a greater risk of mortality (Appendix A). Fasting plasma glucose and serum creatinine were analyzed using dry chemistry methods (Eastman Kodak, Rochester, NY, USA), while serum cystatin C was measured using nephelometry (BN II N, Dade Behring Diagnostic, Marburg, Germany). Estimated glomerular filtration rate (eGFR) was calculated by employing the CKD-EPI creatinine-cystatin C equation [15], with creatinine measurements reassessed between 2010 and 2012 using an IDMS-traceable enzymatic assay for standardization. Multimorbidity was defined as the presence of two or more of the following chronic diseases at baseline: hypertension, cardiovascular disease (CVD), chronic kidney disease (CKD), chronic respiratory disease (CRD) (including asthma and chronic obstructive pulmonary disease), T2D, and cancer. These conditions were selected based on their health impact in older populations [16] and their availability within the PREVEND cohort, drawing from a recommended core list of 20 conditions for multimorbidity research [17]. Multimorbidity status was categorized as no disease, one disease, or multimorbidity (two or more diseases).

### 2.3. Outcome Ascertainment

Information on all-cause mortality was collected from municipality records. All deaths recorded from study enrollment up to 2017 were included in the analysis.

### 2.4. Statistical Analysis

The distribution of continuous variables was assessed visually using histograms and Q-Q plots to evaluate normality. Baseline characteristics were summarized as means (standard deviations, SD) or medians (interquartile ranges, IQR) for continuous variables and as counts with percentages for categorical variables. Cox proportional hazards regression models were used to estimate hazard ratios (HRs) with 95% confidence intervals (CIs) for the associations of MVX, IVX, and MMX with all-cause mortality risk, after confirmation of no major violations of the proportionality of hazards assumptions [18]. Given evidence of graded dose-relationships between the exposures and the risk of all-cause mortality, MVX, IVX, and MMX were modeled as continuous (per 1 SD increment) variables. Confounder adjustments were made based on two models: (Model 1) age and sex and (Model 2) Model 1 plus smoking, alcohol intake, BMI, total cholesterol, HDL-C, triglycerides, and eGFR. The selection of confounders was guided by their established roles as risk factors for mortality in previous studies including the PREVEND study [13] and observed associations in the dataset [19]. All variables included in the analysis had less than 6% missing data (Appendix A); therefore, we performed complete-case analysis without imputation, which is considered appropriate under the assumption of missing at random. Given that IVX and MMX are subcomponents of MVX, the exposures were modeled in separate regression models. Interaction tests were performed to evaluate whether the associations of MVX, IVX, and MMX with mortality risk were modified by multimorbidity status. We also assessed the associations of MVX, IVX, and MMX with mortality across multimorbidity categories by age (using median cut-offs). To compare the associations of MVX, IVX, and MMX with mortality across multimorbidity categories in males and females, we assessed whether these associations differed by sex using HR ratios. The ratio of hazard ratios (RHRs) was computed by dividing the HR for males by the HR for females within each multimorbidity category (no disease, one disease, and multimorbidity). This approach allows for a direct comparison of effect sizes between sexes, providing insight into whether the associations of MVX, IVX, and MMX with mortality are stronger or weaker in males compared to females. An RHR greater than 1 suggests a stronger association in males, while a ratio less than 1 indicates a stronger association in females. Statistical analyses employed R (version 4.0.4, R Foundation for Statistical Computing, Vienna, Austria) and Stata MP version 18 (StataCorp, College Station, TX, USA).

## 3. Results

### 3.1. Baseline Characteristics

Baseline demographics and descriptive characteristics of the 6054 study participants as a whole and categorized by multimorbidity status are presented in Table 1. The mean age of participants at study start was 54 (SD ± 12) years, and 49.9% were males. The three most prevalent comorbidities were hypertension (33.6%), CKD (16.0%), and CRD (9.1%). As the number of morbidities increased, participants tended to be older and had higher levels of MVX, IVX, and MMX. Participants with multimorbidity were more likely to be male and former smokers; have higher levels of BMI, blood pressure, and triglycerides; and have lower HDL-C and eGFR.

### 3.2. Associations of MVX, IVX, and MMX with All-Cause Mortality Risk by Multimorbidity Status

During a median follow-up of 14.0 (IQR, 10.8–14.7) years, which corresponds to 75,393.6 person-years at risk, 911 deaths (annual rate 12.1/1000 person-years at risk; 95% CI: 11.3–12.9) were recorded. Table 2 shows the associations of MVX, IVX, and MMX with the risk of mortality according to multimorbidity status. In analyses adjusted for age, sex, alcohol intake, smoking status, BMI, total cholesterol, HDL-C, triglycerides, and eGFR, the HRs (95% CIs) of mortality per 1 SD increment in MVX were 1.32 (1.13–1.54), 1.23 (1.08–1.40), and 1.29 (1.16–1.43) for individuals with no disease, one disease, and multimorbidity, respectively. The corresponding HRs (95% CIs) were 1.22 (1.05–1.42), 1.17 (1.03–1.34), and 1.25 (1.13–1.38) for IVX and 1.29 (1.11–1.48), 1.16 (1.02–1.31), and 1.14 (1.03–1.25) for MMX. The associations of MVX, IVX, and MMX with the risk of mortality were not significantly modified by multimorbidity status (*p*-values for interactions all >0.05) (Table 2).

The associations were similar in age subgroups (Appendix A). Figure 1 shows the associations of MVX, IVX, and MMX with mortality in each multimorbidity category in males and females. The associations appeared to be stronger in males, particularly among those with existing chronic disease, whereas in females, the associations appeared to be more pronounced in those without chronic conditions. However, estimation of RHRs showed no statistically significant evidence that sex modified the associations of MVX, IVX, and MMX with mortality risk in each multimorbidity category (Table 3).

## 4. Discussion

In this prospective cohort study, our findings highlight that multimorbidity was common in the study population, with hypertension (33.6%), CKD (16.0%), and CRD (9.1%) being the most prevalent conditions. As the number of comorbidities increased, participants tended to be older and exhibited higher levels of MVX, IVX, and MMX, along with adverse cardiometabolic risk profiles. Higher MVX levels were consistently associated with increased mortality risk across all groups, with slightly stronger effects observed in individuals with no disease (32% higher risk) and multimorbidity (29% higher risk). Similarly, IVX was significantly associated with mortality in individuals with no disease (22% higher risk) and multimorbidity (25% higher risk), while MMX showed significant associations across all groups, with the weakest effect in individuals with multimorbidity. However, interaction analyses showed that multimorbidity status did not significantly modify the associations of MVX, IVX, and MMX with mortality. These results indicate that MVX, IVX, and MMX are independent risk indicators of mortality, regardless of the burden of chronic disease, with MVX demonstrating the most robust associations. There was no significant evidence that sex modified the associations of MVX, IVX, or MMX with mortality across multimorbidity categories. Similarly, the associations across age subgroups appeared broadly consistent, although estimates for the younger age group were less precise due to limited sample sizes within each multimorbidity category, which may have reduced statistical power.

While the MVX, IVX, and MMX indices were originally developed and validated by Otvos and colleagues in cohorts of cardiac catheterization patients at high risk for CVD [7], subsequent research has extended their relevance to other clinical populations. For example, these multimarkers have shown associations with neurodegeneration and disability in individuals with multiple sclerosis, supporting their role as potential biomarkers of disease progression beyond cardiovascular contexts [20]. Similarly, elevated MVX, IVX, and MMX scores have been observed in liver transplant recipients and linked to metabolic dysfunction-associated steatotic liver disease, indicating their applicability in transplant and hepatic medicine [21]. Building on this growing body of evidence, our study is the first to evaluate these multimarkers in a large, general population-based cohort, which includes individuals across the spectrum of chronic disease burden—from those with no diagnosed conditions to those with established multimorbidity. This extension enhances the generalizability and population health relevance of MVX, IVX, and MMX. A key novel aspect of our study is the stratification of associations by multimorbidity status, which had not been previously explored. Although Otvos et al. [7] reported consistent associations with mortality regardless of comorbidities, our findings suggest that MVX and IVX were most strongly associated with mortality in individuals with no disease as well as those with multimorbidity, while MMX showed comparatively weaker associations in the multimorbid group. Nevertheless, interaction analyses did not show statistically significant modification of the associations by multimorbidity status, reinforcing the idea that these multimarkers are independent risk indicators of mortality across the spectrum of chronic disease burden. In addition, we evaluated sex- and age-specific associations, formally testing for effect modification, which was not addressed in the original validation studies. Although our findings did not reveal statistically significant sex- or age-based differences, this analytical approach broadens the understanding of these indices’ utility across subpopulations. These findings align with a broader body of research demonstrating the central role of systemic inflammation and metabolic malnutrition in mortality risk, especially in individuals with chronic diseases [8]. Prior studies on inflammatory biomarkers such as *C*-reactive protein (CRP), interleukin-6 (IL-6), and GlycA have consistently shown that chronic low-grade inflammation contributes to adverse health outcomes and increased mortality risk, particularly among those with multimorbidity [22,23]. Similarly, research on malnutrition-related indices has underscored the impact of nutrient and metabolic deficiencies on long-term survival [24,25].

The observed associations between MVX, IVX, and MMX with all-cause mortality likely reflect the complex interplay between systemic inflammation, metabolic malnutrition, and chronic disease progression [26]. MVX, as an integrative measure of these factors, captures underlying physiological vulnerabilities that contribute to mortality risk [7,27]. The slightly stronger associations noted in individuals with no disease and those with multimorbidity suggest that even in the absence of diagnosed chronic conditions, metabolic and inflammatory disturbances may serve as early indicators of mortality risk, while in multimorbidity, these disturbances may exacerbate existing disease pathways and reduce physiological resilience. The IVX-mortality association, particularly in those with chronic disease, aligns with existing evidence that inflammatory pathways, marked by elevated GlycA and reduced small HDL particles, drive endothelial dysfunction, immune dysregulation, and organ damage [28,29]. Inflammation is a key factor in multimorbidity and has been implicated in the pathogenesis of CVD, renal dysfunction, and metabolic disorders [30]—all of which increase mortality risk. The MMX-mortality association, albeit weaker, suggests that protein-energy malnutrition and metabolic inefficiency also play a role in mortality, likely through pathways involving sarcopenia, cachexia, and impaired metabolic adaptation [26,27]. These mechanisms are particularly relevant in multimorbidity, where nutrient deficiencies and muscle wasting contribute to frailty and reduced survival [31]. Although the sex-by-biomarker interaction terms were not statistically significant, the observed patterns are noteworthy. Specifically, the associations of MVX, IVX, and MMX with mortality appeared to be stronger in males with existing chronic conditions, whereas in females, the associations were more pronounced among those without chronic disease. These sex-specific trends may reflect underlying biological differences in immune and metabolic responses and warrant further investigation in future studies to determine whether these patterns represent true sex-based heterogeneity or are attributable to chance.

The findings from this study highlight the clinical relevance of MVX, IVX, and MMX as potential risk stratification tools for mortality, particularly in individuals with chronic disease. Given that metabolic vulnerability and inflammation contribute to mortality risk across all levels of multimorbidity, these biomarkers could help identify high-risk individuals who may benefit from early interventions targeting inflammation and metabolic health. From a public health perspective, addressing metabolic malnutrition and chronic low-grade inflammation through lifestyle modifications, nutritional support, and anti-inflammatory therapies may help mitigate mortality risk in multimorbid populations. Future research should explore whether interventions targeting metabolic vulnerability can improve long-term survival, particularly in those at higher risk due to chronic disease burden. Despite their promising utility, the practical implementation of MVX, IVX, and MMX in routine clinical care faces challenges. These indices are derived using NMR spectroscopy and proprietary algorithms, which are currently available in specialized clinical laboratories in the USA and Canada but less widely available in other regions. However, as NMR-based technologies gain traction in clinical settings—particularly in Europe, where clinical NMR is expanding—the feasibility of integrating these indices into risk prediction models will likely increase.

Importantly, these multimarkers may complement rather than replace existing risk scores by capturing latent metabolic and inflammatory processes not reflected in traditional clinical variables. Future work should focus on validating these indices in diverse populations, assessing cost-effectiveness, and exploring simplified surrogate panels that approximate MVX, IVX, and MMX using more readily available biomarkers. Such advancements could facilitate broader adoption and enhance personalized risk assessment and prevention strategies in both primary and secondary care.

## 5. Strengths and Limitations

This study is the first to prospectively examine the associations of MVX, IVX, and MMX with all-cause mortality while accounting for multimorbidity status, providing potential insights into the role of metabolic vulnerability and inflammation in mortality risk stratification. The large sample size and extended follow-up enhance the statistical power and reliability of the findings. Additionally, multimorbidity was comprehensively assessed using six prevalent chronic diseases, and the analysis was adjusted for a broad range of confounders, improving the robustness of the results. However, some limitations must be considered. As noted in previous research by Otvos et al. [7], the equations used to derive MVX and its subcomponents should be interpreted with caution, as some biomarker measurements, such as branched-chain amino acids, may have limitations. Furthermore, certain prevalent chronic conditions—such as depression, neurodegenerative diseases (e.g., Alzheimer’s and Parkinson’s disease), and clinical nutritional disorders—were not included in the multimorbidity definition due to data limitations within the cohort. The omission of these conditions may have led to an underestimation of the true prevalence of multimorbidity, particularly among older adults, where such diagnoses are common. This restricted condition set may also have resulted in misclassification of disease burden, potentially attenuating the observed associations between multimorbidity status and mortality risk. Additionally, these exclusions limit the ability to fully capture the complex interplay between physiological, psychological, and metabolic factors that contribute to vulnerability and adverse outcomes. Other inherent limitations of observational studies apply, including measurement error, residual confounding, and the inability to establish causality. Additionally, biomarker levels were assessed only at baseline, limiting insights into how dynamic changes over time may influence mortality risk. Another limitation is the absence of direct dietary or clinical nutritional data, such as food intake, anthropometric indicators, or validated malnutrition screening tools. As a result, the MMX, which is derived from circulating biomarkers of protein-energy wasting and metabolic inefficiency, should be interpreted as a proxy indicator of metabolic malnutrition rather than a direct measure of clinically assessed nutritional status. While MMX may capture metabolic signatures linked to malnutrition-related processes such as sarcopenia and frailty, it does not reflect nutritional intake or malnutrition diagnoses based on established clinical criteria. This limits its comparability to traditional nutritional assessments and may restrict its applicability in clinical nutrition settings. Finally, as the study population was predominantly Caucasian, generalizability to more diverse racial and ethnic groups is limited, warranting further validation in broader populations.

## 6. Conclusions

This study provides novel evidence that MVX, IVX, and MMX are independent risk indicators of all-cause mortality across different multimorbidity statuses, with MVX showing the strongest associations. The findings highlight the role of metabolic vulnerability and inflammation in shaping mortality risk, regardless of the burden of chronic disease. However, it is important to emphasize that MMX should be viewed as a proxy measure of metabolic malnutrition, derived from circulating metabolites, and not as a direct or validated clinical nutritional assessment tool. This distinction is essential for interpretation and highlights the need for future research to integrate MMX with traditional dietary and anthropometric measures to better characterize nutritional risk.

## Figures and Tables

**Figure 1 nutrients-17-01747-f001:**
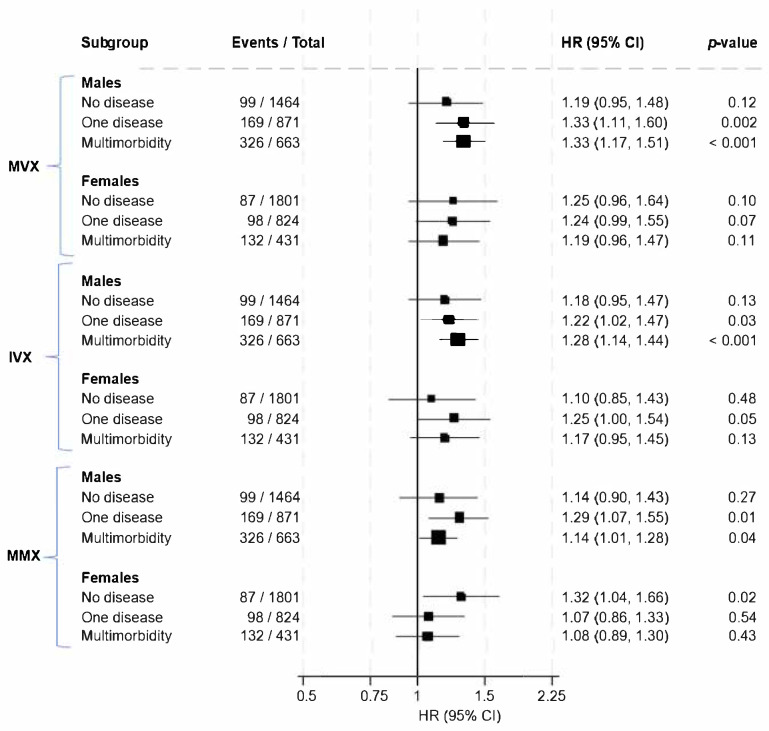
Associations of MVX, IVX, and MMX with mortality according to multimorbidity status and sex. HRs are per 1 standard deviation increase in each exposure. CI, confidence interval; HR, hazard ratio; IVX, inflammation vulnerability index; MMX, metabolic malnutrition index; MVX, metabolic vulnerability index. Models were adjusted for age, smoking, alcohol intake, body mass index, total cholesterol, high-density lipoprotein cholesterol, triglycerides, and estimated glomerular filtration rate.

**Table 1 nutrients-17-01747-t001:** Baseline characteristics of participants overall and by status at end of follow-up. Continuous variables are reported as mean (SD) or median (interquartile range), and categorical variables are reported as n (%).

		Baseline Disease Status			
Variable	Overall (N = 6054) Mean (SD) or Median (IQR)	No Disease (N = 3265) Mean (SD) or Median (IQR)	One Disease (N = 1695) Mean (SD) or Median (IQR)	Multimorbidity (N = 1094) Mean (SD) or Median (IQR)	*p*-Value
MVX	44.9 (9.0)	43.7 (8.9)	45.5 (8.7)	48.0 (9.0)	<0.001
IVX	39.3 (10.5)	37.5 (10.2)	39.9 (10.3)	43.5 (10.5)	<0.001
MMX	54.3 (6.4)	54.2 (6.4)	54.4 (6.2)	54.4 (6.6)	0.47
GlycA (µmol/L)	373 (337, 416)	358 (325, 399)	383 (348, 422)	402 (365, 443)	<0.001
Small HDL particles (µmol/L)	15.90 (2.43)	15.93 (2.40)	16.01 (2.45)	15.65 (2.47)	<0.001
Leucine (µmol/L)	104.18 (27.02)	100.94 (25.14)	105.31 (27.80)	112.09 (29.36)	<0.001
Valine (µmol/L)	201.29 (38.13)	194.44 (35.69)	205.03 (39.04)	215.95 (38.84)	<0.001
Isoleucine (µmol/L)	58.28 (14.56)	55.66 (13.31)	59.85 (15.09)	63.66 (15.47)	<0.001
Citrate (µmol/L)	2.09 (0.48)	2.02 (0.45)	2.12 (0.46)	2.24 (0.52)	<0.001
**Questionnaire**					
Age (years)	54 (12)	48 (10)	57 (11)	64 (10)	<0.001
Males, n (%)	2998 (49.5)	1464 (44.8)	871 (51.4)	663 (60.6)	<0.001
Alcohol consumers, n (%)	4528 (74.8)	2565 (78.6)	1215 (71.7)	748 (68.4)	<0.001
Smokers, n (%)					<0.001
Never smokers	1744 (28.8)	1062 (32.5)	461 (27.2)	221 (20.2)	
Former smokers	2618 (43.2)	1241 (38.0)	783 (46.2)	594 (54.3)	
Light current smokers	649 (10.7)	370 (11.3)	166 (9.8)	113 (10.3)	
Heavy current smokers	1043 (17.2)	592 (18.1)	285 (16.8)	166 (15.2)	
**Chronic diseases**					
History of T2D, n (%)	367 (6.1)	0 (0.0)	77 (4.5)	290 (26.5)	<0.001
History of CVD, n (%)	180 (3.0)	0 (0.0)	19 (1.1)	161 (14.7)	<0.001
History of CKD, n (%)	971 (16.0)	0 (0.0)	267 (15.8)	704 (64.4)	<0.001
History of hypertension, n (%)	2037 (33.6)	0 (0.0)	1038 (61.2)	999 (91.3)	<0.001
History of CRD, n (%)	553 (9.1)	0 (0.0)	261 (15.4)	292 (26.7)	<0.001
History of cancer, n (%)	111 (1.8)	0 (0.0)	33 (1.9)	78 (7.1)	<0.001
**Physical measurements**					
BMI (kg/m^2^)	26.7 (4.4)	25.5 (3.7)	27.5 (4.4)	29.0 (4.9)	<0.001
SBP (mmHg)	126 (19)	117 (11)	133 (19)	142 (21)	<0.001
DBP (mmHg)	73 (9)	70 (7)	76 (9)	78 (10)	<0.001
**Lipid and renal markers**					
Total cholesterol (mmol/L)	5.43 (1.05)	5.35 (1.02)	5.58 (1.06)	5.43 (1.10)	<0.001
HDL-C (mmol/L)	1.26 (0.31)	1.30 (0.31)	1.24 (0.30)	1.17 (0.31)	<0.001
Triglycerides (mmol/L)	1.12 (0.81, 1.61)	1.00 (0.74, 1.41)	1.22 (0.90, 1.76)	1.39 (1.02, 1.91)	<0.001
Creatinine (mg/dl)	0.83 (0.24)	0.80 (0.13)	0.81 (0.16)	0.93 (0.45)	<0.001
Cystatin C (mg/L)	0.91 (0.21)	0.85 (0.13)	0.91 (0.16)	1.08 (0.36)	<0.001
Estimated GFR (mL/min/1.73 m^2^)	84.0 (10.6)	87.4 (5.6)	83.5 (9.1)	74.5 (16.7)	<0.001

Abbreviations: BMI, body mass index; CKD, chronic kidney disease; CRD, chronic respiratory disease; CVD, cardiovascular disease; DBP, diastolic blood pressure; GFR, glomerular filtration rate (as calculated using the Chronic Kidney Disease Epidemiology Collaboration combined creatinine–cystatin C equation); HDL-C, high density lipoprotein cholesterol; IQR, interquartile range; IVX, the inflammation vulnerability index; MMX, the metabolic malnutrition index; MVX, the metabolic vulnerability index; SBP, systolic blood pressure; SD, standard deviation; T2D, type 2 diabetes. Former smokers were those who were non-smokers at the time of study inclusion but had ever smoked in their life; current smokers were those who reported smoking at the time of inclusion; light current smokers were current smokers who reported smoking 10 cigarettes or less per day; and heavy current smokers were current smokers who reported smoking more than 10 cigarettes per day.

**Table 2 nutrients-17-01747-t002:** Associations of MVX, IVX and MMX with all-cause mortality among individuals with and without multiple chronic diseases.

	No Disease					One Disease					Multimorbidity					*p*-Value *
	Events/Total	Model 1 HR (95% CI)	*p*-Value	Model 2 HR (95% CI)	*p*-Value	Events/Total	Model 1 HR (95% CI)	*p*-Value	Model 2 HR (95% CI)	*p*-Value	Events/Total	Model 1 HR (95% CI)	*p*-Value	Model 2 HR (95% CI)	*p*-Value	
MVX	186/3265	1.40 (1.20–1.63)	<0.001	1.32 (1.13–1.54)	<0.001	267/1695	1.28 (1.12–1.45)	<0.001	1.23 (1.08–1.40)	0.002	458/1094	1.35 (1.22–1.49)	<0.001	1.29 (1.16–1.43)	<0.001	0.74
IVX	186/3265	1.29 (1.08–1.39)	0.001	1.22 (1.05–1.42)	0.010	267/1695	1.23 (1.08–1.39)	0.002	1.17 (1.03–1.34)	0.015	458/1094	1.31 (1.19–1.45)	<0.001	1.25 (1.13–1.38)	<0.001	0.74
MMX	186/3265	1.28 (1.12–1.48)	<0.001	1.29 (1.11–1.48)	0.001	267/1695	1.15 (1.01–1.30)	0.032	1.16 (1.02–1.31)	0.024	458/1094	1.15 (1.05–1.26)	0.004	1.14 (1.03–1.25)	0.009	0.33

CI, confidence interval; HR, hazard ratio; IVX, the inflammation vulnerability index; MMX, the metabolic malnutrition index; MVX, the metabolic vulnerability index. HRs are per 1 standard deviation increase. Model 1: Age and sex. Model 2: Model 1 plus smoking, alcohol intake, body mass index, total cholesterol, high-density lipoprotein cholesterol, triglycerides, and estimated glomerular filtration rate. *, *p*-values for interaction are for model 2 and are for the effect of multimorbidity status on the associations of MVX, IVX, and MMX with risk of mortality.

**Table 3 nutrients-17-01747-t003:** Ratio of hazard ratios comparing males to females for the associations of MVX, IVX, and MMX with all-cause mortality across multimorbidity categories.

	No Disease			One Disease			Multimorbidity		
	Males Events/Total	Females Events/Total	RHR (Males/Females) (95% CI)	Males Events/Total	Females Events/Total	RHR (Males/Females) (95% CI)	Males Events/Total	Females Events/Total	RHR (Males/Females) (95% CI)
MVX	99/1464	87/1801	0.95 (0.67–1.34)	169/871	98/824	1.08 (0.81–1.44)	326/663	132/431	1.12 (0.87–1.43)
IVX	99/1464	87/1801	1.08 (0.76–1.51)	169/871	98/824	0.98 (0.74–1.31)	326/663	132/431	1.09 (0.86–1.39)
MMX	99/1464	87/1801	0.86 (0.62–1.20)	169/871	98/824	1.20 (0.91–1.60)	326/663	132/431	1.05 (0.84–1.32)

CI, confidence interval; IVX, the inflammation vulnerability index; MMX, the metabolic malnutrition index; MVX, the metabolic vulnerability index; RHR, ratio of hazard ratio. Analysis adjusted for age, systolic blood pressure, smoking, alcohol intake, body mass index, total cholesterol, high-density lipoprotein cholesterol, triglycerides, and estimated glomerular filtration rate. HRs are per 1 standard deviation increase.

## Data Availability

The data can only be accessed on reasonable request from the Principal Investigators.

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
