# Peer review of "Associations of Multimarkers of Metabolic Malnutrition and Inflammation with All-Cause Mortality by Multimorbidity Status"

_nutrients, 2025, doi:10.3390/nu17111747_

Round 1

Reviewer 1 Report

Comments and Suggestions for Authors

A very well written article on metaboli and inflamation association with multimorbidity

The title and abstract is well presented

The method- well decribed and presented

English grammar- no issues detected

Results: good presentation with good tables and figures.

The discussion- well conected with the results

Conclussion: short and concise.

No major issues detected, no flaws in the ethods or results

Author Response

Thank you for reviewing our manuscript and for positive comments.

Reviewer 2 Report

Comments and Suggestions for Authors

Dear Authors,

I like the idea of your research. As the novelty is clear, the methodology is well-described, the one thing I can suggest is to enhance the discussion section.  More references to the previous studies would be welcome.

Best regards,

The reviewer.

Author Response

Thank you for your positive feedback and for recognizing the novelty and clarity of our methodology. We appreciate your suggestion to strengthen the discussion section. In response, we have revised the discussion to include additional references to relevant studies, particularly those exploring the roles of systemic inflammation, metabolic malnutrition, and multimorbidity in shaping related outcomes.

Reviewer 3 Report

Comments and Suggestions for Authors This manuscript explores the association between the MVX and its subcomponents IVX and MMX with all-cause mortality, stratified by multimorbidity status, in the PREVEND cohort.  Main comments 1. The authors interpret malnutrition on the basis of metabolic biomarkers (MMX), without any direct nutritional assessment (e.g. food intake, anthropometric indicators or clinical malnutrition screening tools). Given the objective of the journal, this represents a significant limitation. Authors should explicitly state that no dietary or clinical nutritional data were collected or analysed and discuss the implications for the interpretation of MMX as an indicator of malnutrition. 2.The MVX, IVX and MMX indices were previously developed and validated by Otvos et al. This study extends their use to a general population cohort stratified by multimorbidity status. However, the novelty of the current contribution is somewhat limited. The authors should better highlight what is new compared to Otvos et al. in addition to the population change. 3.The use of NMR spectroscopy and proprietary algorithms limits the clinical applicability of these indices due to high cost and lack of accessibility. A comment on practical implementation barriers should be added to the discussion. 4.Analysis of gender interactions is appropriate, but the results (although not statistically significant) show interesting trends (e.g. stronger associations of IVX in women without chronic conditions). The authors should briefly discuss these non-significant patterns, as they could be a source of hypotheses. 5. The selected chronic conditions exclude important categories such as neurodegenerative diseases, depression and nutritional disorders. Authors should recognise this limitation, as it could lead to an underestimation of the prevalence of multimorbidity and distorted associations. 6. Although multivariable models are appropriate, there is no mention of how missing data were handled or whether collinearity between IVX and MMX was assessed. Please clarify both points in the Methods section. 7. There is plagiarism between the methods 8. There are too many self-citations   Minor comments The manuscript is generally well written, although the Discussion section is repetitive in some places (e.g. the strength of MVX as a predictor of mortality is repeated too much). Table 1: Indicate whether normality of distributions has been tested for continuous variables. The article would benefit from a graphical abstract summarising the MVX-mortality association according to multimorbidity. Ensure that supplementary figures are cited in order (figure S2 is cited before S1).

Round 2

Reviewer 3 Report

Comments and Suggestions for Authors   Thank you for your careful review of the manuscript and for your comprehensive responses to the reviewers' comments. The revised version is much improved. The methodology is clearly described and the statistical analysis is appropriate. The integration of MVX, IVX and MMX with multimorbidity status is well conducted and offers valuable insights.   I have only a few final suggestions: 1.While acknowledging the lack of direct dietary or clinical nutritional data, it might be useful to reiterate in the conclusions that MMX should be considered a proxy for metabolic malnutrition and not a validated nutritional assessment tool. 2.The discussion could benefit from further expansion of the implications for clinical practice and the potential integration of these indices into routine risk stratification, especially considering the current limitations of accessibility. 3.Please check again the order of references to ensure that self-citations are justified and limited to essential ones. 4.In Figure 1 and Table 3, the patterns of association by sex are well described, but a brief comment on these trends in the abstract would strengthen the manuscript.
